# Microglia Signatures: A Cause or Consequence of Microglia-Related Brain Disorders?

**DOI:** 10.3390/ijms252010951

**Published:** 2024-10-11

**Authors:** Alessandra Mirarchi, Elisabetta Albi, Cataldo Arcuri

**Affiliations:** 1Department of Medicine and Surgery, University of Perugia, Piazza L. Severi 1, 06132 Perugia, Italy; alessandra.mirarchi@dottorandi.unipg.it; 2Department of Pharmaceutical Sciences, University of Perugia, Via Fabretti 48, 06123 Perugia, Italy; elisabetta.albi@unipg.it

**Keywords:** microglia signatures, neuroinflammation, multiple sclerosis, neurodegeneration, Alzheimer’s disease, TREM2, APOE, aging

## Abstract

Microglia signatures refer to distinct gene expression profiles or patterns of gene activity that are characteristic of microglia. Advances in gene expression profiling techniques, such as single-cell RNA sequencing, have allowed us to study microglia at a more detailed level and identify unique gene expression patterns that are associated, but not always, with different functional states of these cells. Microglial signatures depend on the developmental stage, brain region, and specific pathological conditions. By studying these signatures, it has been possible to gain insights into the underlying mechanisms of microglial activation and begin to develop targeted therapies to modulate microglia-mediated immune responses in the CNS. Historically, the first two signatures coincide with M1 pro-inflammatory and M2 anti-inflammatory phenotypes. The first one includes upregulation of genes such as CD86, TNF-α, IL-1β, and iNOS, while the second one may involve genes like CD206, Arg1, Chil3, and TGF-β. However, it has long been known that many and more specific phenotypes exist between M1 and M2, likely with corresponding signatures. Here, we discuss specific microglial signatures and their association, if any, with neurodegenerative pathologies and other brain disorders.

## 1. Introduction

Microglia, the brain parenchymal macrophages, play a pivotal role in the physiological function of the central nervous system (CNS) and various neurological diseases, including stroke, brain tumors, Alzheimer’s disease (AD), and amyotrophic lateral sclerosis (ALS) [1,2]. Beyond their conventional roles in immune surveillance and clearing cellular debris, recent research has unveiled their active involvement in shaping neural development. Microglia influence processes like neurogenesis and synaptic pruning, thereby contributing to the fine-tuning of the brain’s architecture [3,4,5,6,7].

Despite their significance as multitasking cells, there is a considerable gap in our knowledge regarding the molecular diversity of microglia under normal physiological conditions. This gap is especially notable during developmental stages when microglia perform vital non-immune functions.

Furthermore, microglia exhibit transcriptomic similarities to other myeloid cells that can infiltrate the brain parenchyma during disease [8,9]. Therefore, it is imperative to undertake a systematic comparison between microglia and these related immune cells to better understand their distinct characteristics and functions, particularly in the context of neurological health and disease.

Microglia, along with many other tissue macrophages, are remarkable in their long-lived, self-renewing nature, originating from erythro-myeloid progenitors in the yolk sac [4,10,11]. In mice, microglia migrate to the brain at around embryonic day 9 (E9), and it has been proposed that the closure of the blood–brain barrier (BBB) around E13.5 serves to confine microglia within the brain parenchyma [12]. Despite this intricate developmental journey, studies employing bulk RNA sequencing (RNA-seq) data have outlined a step-wise differentiation program for microglia [13]. However, these studies often rely on general surface markers, potentially missing out on microglial heterogeneity, including transient populations during development, which could lead to an underestimation of microglia’s developmental complexity.

Furthermore, while it has been observed that mature microglia exhibit non-uniform distribution and distinct morphologies in different brain regions [14], seemingly correlating with region-specific gene expression profiles [15,16,17], it remains an open question whether there are well-defined molecular subtypes of microglia in the adult brain. If such subtypes do exist, their distribution across various brain regions remains an intriguing and unanswered aspect of microglial biology.

## 2. Origin of Microglia

It is now known that microglial cell precursors originate in the yolk sac. In mice, microglial precursor cells emerge between embryonic days E7.0 and E9.0, coinciding with an initial “primitive” wave of hematopoiesis. Significantly, cells expressing hematopoietic and macrophage/microglia markers become detectable in the developing brain from day E9.5 [12] and postnatal hematopoietic progenitors do not play a role in sustaining adult microglial cells [12]. With the exception of choroid plexus macrophages, non-parenchymal macrophages (meningeal and perivascular) originate in the yolk sac and persist throughout life without contributions from bone marrow precursors [8].

Microglial cell development in mice is closely tied to the colony-stimulating factor-1 receptor (CSF-1R), unlike macrophages and circulating monocytes [12]. The nature of microglial precursors and their gene expression patterns are still not fully understood [10,18,19]. The involvement of circulating monocytes in the adult microglial population is confined to certain pathological conditions, likely involving the recruitment of monocytes from the bloodstream due to alterations in blood–brain barrier permeability [20,21,22].

In humans, amoeboid microglial cells enter the cerebral wall from leptomeninges and ventricular lumen starting at 4.5 weeks of gestation [23]. During early brain development, these cells penetrate the cerebral cortex and white matter in an amoeboid form. Subsequently, microglial cells migrate and colonize the parenchyma, acquiring a ramified morphology characteristic of the adult brain, and fulfilling their functions. Specific molecular signals orchestrate all events regarding microglia functions in a temporal and spatial manner.

## 3. Microglia Signature

### 3.1. Region-Specific Gene Expression Promfiles of Microglia

The microglial population within the brain displays non-uniform characteristics. In murine models, microglia exhibit unique transcriptional identities that are dependent on the region [17,24]. Recent progress in single-cell sequencing has unveiled a more detailed understanding of the diversity and regional heterogeneity of murine brain microglia [25,26]. Intriguingly, these findings indicate that microglial diversity is most pronounced in the developing, aged, or injured brain, and utilizing a multiple sclerosis (MS) murine model, but also human MS brain tissues, various activated subpopulations of microglia have been observed in demyelinating lesions [26]. Alternative research groups have supplemented these observations by employing advanced techniques, uncovering comparable findings from the tissue of healthy human brains [27,28].

When comparing microglia from the mouse hippocampus, cerebral cortex, striatum, and cerebellum, there is an observed regional diversity in transcriptional patterns. Three distinct transcriptomic clusters emerge, specific to the cerebellum, hippocampus, and cerebral cortex/striatum. Considering associated biological processes, it was seen that the microglial gene cluster of the hippocampus participated in the regulation of energy production, while the cortical and cerebellar clusters are associated with genes governing the immune response. In line with this, analysis of transcription factor binding motifs reveals an over-representation of transcription factors regulating bioenergetic genes in the hippocampal cluster and those governing immune and inflammatory genes in the cerebellar cluster [17].

It is noteworthy that there appears to be a distinction in the immune activation status of microglia associated with the cortical and cerebellar clusters. Microglia from the cortex exhibit an elevated expression of genes encoding inhibitory immunoreceptors, such as triggering receptors expressed on myeloid cells 2 (Trem2) and SiglecH. Conversely, microglia from the cerebellum demonstrate an upregulation of genes encoding activating immunoreceptors, suggesting a more immune-activated microglial phenotype, although distinct from the phenotypes induced by lipopolysaccharide (LPS) or IL-43. Importantly, about one third of the microglial sensome genes, part of the homeostatic microglial gene signature, exhibit differential expression in microglia derived from distinct brain regions. In conclusion, while microglia across different brain regions share the expression of single specific genes, more importantly, they also show region-specific gene set expression, indicating unique and specific microglial functions associated with each region [17]. To further confirm this, previous works have evidenced distinct microglial phenotypes when comparing regions within the basal ganglia [16]. The transcriptome analysis reveals that microglia in the ventral tegmental area (VTA) exhibit the most significant differences compared to microglia in the substantia nigra pars compacta and nucleus accumbens. In VTA, microglia genes differentially expressed are primarily associated with metabolic processes, including glycolysis, oxidative phosphorylation, and gluconeogenesis. Microglia in the substantia nigra pars compacta and VTA display limited contribution to both homeostasis and immunological surveillance, as evidenced by observations of cell density, lysosome content, and cellular branching. Despite these regional differences, the expression profile of microglia among the various regions evidences the preservation of canonical microglial cell functions.

When microglia were examined alongside macrophages located on the choroid plexus, perivascular spaces, and subdural meninges, an interesting high degree of similarity was evidenced. This similarity did not present with peripheral monocytes. If the monocyte transcriptome was juxtaposed with that of microglia and non-parenchymal macrophages, more than 400 differentially expressed genes were expressed in common, both in parenchymal (microglia) and non-parenchymal macrophages, among these, Iba1, C-X3-C motif chemokine receptor 1 (Cx3cr1), and Csf1r. Beyond this shared gene set, microglia and non-parenchymal macrophages also express distinct, separate gene sets and, in this context, more than 2300 genes have been identified [8].

### 3.2. The Human Homeostatic Microglia Signature (Figure 1)

A recent study identified the gene signature for homeostatic human microglia from a cohort of 39 adult postmortem donors, ranging from 34 to 102 years old [29]. This gene signature comprises more than 1200 genes that exhibit significant differential expression in purified microglia compared to cell lysate using whole parietal cortices. Gene Ontology analysis indicated that these genes are associated with the innate immune system, encompassing functions such as pathogen and self-recognition, inflammasome, cell adhesion, and motility (C3XCR1), as well as immune signaling and modulation (P2RY12, HLA-DR, and C1QA-C). Furthermore, apolipoprotein E (APOE) and TREM2, known to be neurodegenerative disease risk genes, are enriched in microglia purified by the adult human brain. Additionally, the two transcription factors, interferon regulatory factor 8 (IRF8) and PU.1, which also play a fundamental role in murine microglia ontogeny and development, are highly expressed.

In another study [30], the transcriptomes of microglia isolated from seemingly healthy brain tissue acquired during neurosurgery from 19 17-year-old patients with tumors, acute stroke, or epilepsy were analyzed. The top 30 genes highly expressed in that dataset are associated with functions like synaptic remodeling (C1QA-C and C3), microglial motility and ramification (CX3CR1 and P2RY12), and immune response (HLA-DRA and HLA-B). Comparing whole cortex vs. microglia-specific gene expression profiles, more than 800 homeostatic human microglial signature genes were identified, including P2RY12, CX3CR1, and several complement receptors like C1QA, C1QB, C1QC, and C3. Moreover, these homeostatic signature genes in human microglia significantly coincide with transcriptomic datasets linked to various neurological diseases, including AD and Parkinson’s disease (PD), where many of the homeostatic microglial signature genes show differential expression. This indicates a crucial role of microglia in the pathophysiology of these diseases [31].

**Figure 1 ijms-25-10951-f001:**
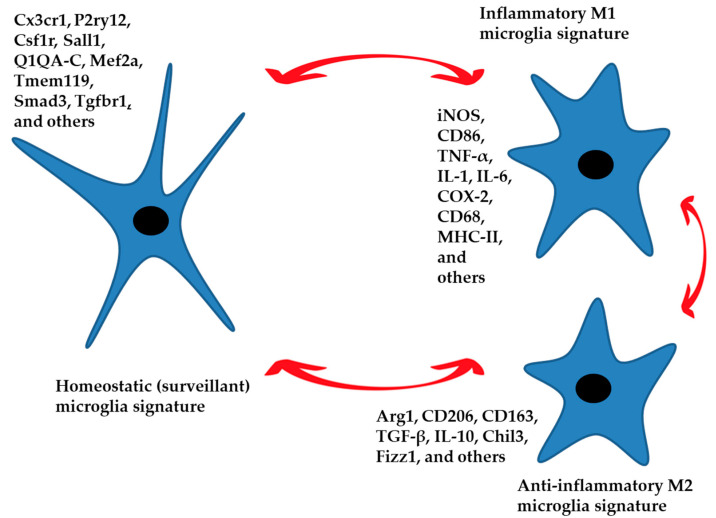
The homeostatic signature of microglia performs control and surveillance functions of the nervous parenchyma. These functions are region-specific. An inflammatory phenotype in response to specific external stimuli can be established, generating a defensive response. The anti-inflammatory phenotype leads to the resolution of the damage by limiting the inflammatory response and returning the microglia to a homeostatic state of surveillance. Aging and neurodegenerative disease can also generate an inflammatory phenotype. The three phenotypes represented in the figure are characterized by specific signatures with numerous intermediate states between M1 and M2 that push the microglia toward a more or less inflammatory or anti-inflammatory function. Gene expression markers for each condition are listed.

The homeostatic gene signature of microglia exhibits conservation across species [31,32]. When comparing the two distinct gene signatures characterizing homeostatic human microglia with several signatures of homeostatic murine microglia, an overlap of more than 50% is evident, contingent on the specific datasets utilized for comparison [31,32]. Genes such as APOC1, MPZL1, SORL1, CD58, ERAP2, GNLY, and S100A12, closely linked to the innate immune system, are uniquely expressed in human microglia and either absent or expressed to a minimal extent in murine microglia. The substantial congruence between murine and human transcriptomes aligns with the identification of analogous epigenetic landscapes and specifically microglial-specific regulatory regions [32]. To sum up, recent research has delineated gene signatures distinctive to homeostatic microglia in both mice and humans, laying the groundwork for enhanced identification and exploration of microglia in murine and human tissues.

### 3.3. M1 Microglia Signature

The M1 microglia signature plays a pivotal role in the host’s defense mechanisms, serving as potent effector cells adept at eliminating intracellular microorganisms and tumor cells (Figure 1). Moreover, they unleash substantial amounts of pro-inflammatory cytokines, contributing to inadvertent damage in healthy tissues. M1 microglia are characterized by their production of elevated levels of oxidative metabolites and pro-inflammatory cytokines, such as tumor necrosis factor-α (TNF-α), IL-6, IL-1β, and IL-12 [33]. The persistent release of pro-inflammatory cytokines, reactive oxygen species (ROS), and unregulated glutamate release may exert neurotoxic effects, although the question of microglial-induced neurotoxicity remains an open one. Certain authors [34] posit that microglial cells play a beneficial role in resolving numerous CNS diseases and they argue that microglia become deleterious due to diminished expression of crucial receptors (CD200R, CX3CR1, and TREM2) or to mutations, particularly during instances of aging and neurodegenerative diseases; essentially, this effect occurs in scenarios where microglia deviate from their innate properties and become either tolerogenic or hyperactive [35,36].

In elderly animals, there is a reduction of the microglia mobility and of the branching processes of cells. This decrease may result in less effective surveillance and potentially weakened defense against tissue damage. Furthermore, in vitro evidence suggests that in the aging brain, microglia exhibit an increased expression of the major histocompatibility complex (MHC) class II molecules and become less responsive to regulatory cues such as colony-stimulating factor 1 (CSF1) or transforming growth factor (TGF)-β1 [37]. Over the course of their lifespan, instances of cytokine stimulation and systemic inflammation can instruct microglia, heightening their reactivity [38]. The process of encountering various harmful stimuli is termed priming. In conjunction with this priming, the damage of DNA and the gradual accumulation of mutations due to aging can cause microglia to develop an increasingly hypersensitive phenotype. This results in heightened immune responsiveness and a greater resistance to regulatory mechanisms [37].

In the realm of neurodegenerative diseases, the activation of microglial toll-like receptors (TLRs) by components of deceased cells can instigate the release of pro-inflammatory mediators, including IL-1β, TNF-α, nitric oxide (NO), ROS, and IL-6. These mediators, in a cascading effect, have the potential to expedite neuronal degeneration. Notably, the inflammation set in motion by endogenous molecules, such as protein aggregates and remnants of deceased neurons, may serve as a catalyst for the progression of neurodegenerative disorders such as MS and AD [39]. However, the multifaceted nature of microglial effects is evident. On the one hand, in the context of MS, microglia actively contribute to neurodegenerative and neuroinflammatory processes by releasing inflammatory mediators and facilitating the activity and infiltration of leukocytes into the CNS. Conversely, microglia also play a vital role in CNS repair by generating neurotrophic factors and clearing inhibitory myelin debris. Additionally, there is evidence suggesting that M1 microglia may exert a positive influence on the regulation of neurogenesis through the secretion of neurotrophic mediators such as TGF-β and insulin-like growth factor (IGF)-1 [40].

### 3.4. M2 Microglia Signature

M2 macrophages, occasionally further classified into M2a, M2b, and M2c subtypes, aid in wound repair and mitigate harmful immune responses [41] (Figure 1). Whether this nomenclature may be associated with microglia is debated. However, the M2 microglial phenotype enhances neuron longevity and minimizes brain injury by expressing high levels of chitinase 3-like 3 (Chil3) and arginase-1 (Arg-1), which prevent extracellular matrix breakdown, found in the inflammatory zone, (FIZZ)1, which supports extracellular matrix formation. Moreover, M2 microglia increase IL-10, TGF-β, and IGF-1 production [42]. The enzyme Arg-1 transforms arginine into ornithine, facilitating wound healing. By utilizing arginine, which is also the substrate for inducible nitric oxide synthase (iNOS), Arg-1 can effectively compete with iNOS, thereby reducing nitric oxide production [43]. Consequently, iNOS and Arg-1 serve as clear indicators for distinguishing M1 and M2 phenotypes, respectively.

Another way to describe the functions and phenotypes is by the cytokines that trigger this phenotype, i.e., M2 [42]. In this context, not all cytokines that define the M2 phenotype in macrophages are effective in microglia and some M2 markers found in macrophages are not expressed in the microglial cells.

CD206, also known as the mannose receptor, is commonly used as a marker to characterize M2 microglia, a subset of activated microglia with anti-inflammatory and tissue repair functions [44]. However, identifying exclusive markers for M2 microglia, not expressed in M2 macrophages, is challenging due to the shared phenotypic traits and molecular markers between these two cell types, stemming from their common origin as innate immune cells. A potential marker specific to microglia (both M1 and M2) compared to macrophages is TMEM119. This marker has been identified as specific to resident microglia in the brain and is not generally expressed by peripheral macrophages [45]. A marker specific to M2 macrophages that is not typically expressed by M2 microglia is CD163. CD163 is a surface receptor present on M2 macrophages and is used as a distinctive marker for these cells [46].

Of course, due to the distinct origins and responses of microglia and macrophages, their roles in either alleviating or exacerbating pathology may also differ.

Chen and Trapp (2015) recently identified microglia as crucial defenders of the CNS during development, adulthood, and disease [47]. They describe how microglial activities are adjusted as needed, allowing microglia to transition from the M1 to the M2 phenotype, with the latter playing a vital role in promoting recovery [47]. Normally, immune responses are meticulously regulated to initiate and resolve appropriately, maintaining tissue homeostasis. However, under pathological conditions, these immune responses become dysregulated, leading to an imbalance that skews towards either excessive inflammation or suppression.

Another study suggested that glatiramer acetate could stimulate microglial cells to express IGF-1 for neuroprotective action [48]. A study on experimental autoimmune encephalomyelitis (EAE) in a mouse model of MS [49], reported increased expression of chitinase-like 3 (Ym1), regulated by CNS-derived IL-4. In agreement with this, clinical signs of EAE were aggravated in chimera mice lacking IL-4, probably due to a correlation with a reduced M2 microglial phenotype [47].

Kigerl et al. (2009) studied the effects of conditioned media from bone marrow-derived macrophages on the survival and neurite outgrowth of dorsal root ganglion cells in a mouse spinal cord injury model [50]. They found that conditioned media from macrophages activated with interferon-γ and LPS harmed neurons, whereas media from macrophages activated with IL-4 promoted neurite outgrowth. These data may also suggest that the polarization of resident microglia towards an activated M2 phenotype could promote central nervous system repair by limiting inflammation-mediated secondary damage. 

### 3.5. Microglia and Aging (Figure 2)

In advanced age, there is a persistent, low-level inflammation in the brain characterized by increased expression of MHC-II, complement receptors, and pro-inflammatory cytokines, alongside a decrease in anti-inflammatory gene activity [51]. Moreover, there is an accumulation of reactive oxygen/nitrogen species and displaced self-molecules in the brain, which affects the functioning of microglia. Even under normal conditions, aged microglia tend to produce higher levels of cytokines [52], but when exposed to inflammatory triggers, aged microglia exhibit heightened responsiveness, producing larger quantities of cytokines over extended periods [53]. This heightened inflammatory response likely contributes to the prolonged illness experienced by elderly individuals during sickness or infections, often accompanied by symptoms like cognitive decline and/or depression [38]. Moreover, this chronic inflammation may contribute to the development of age-related neurodegenerative diseases [54].

**Figure 2 ijms-25-10951-f002:**
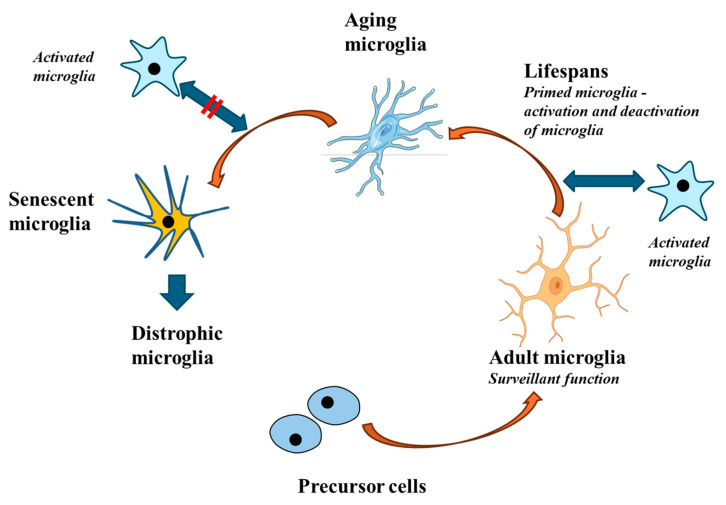
During aging, microglial responses become less intense. While young and adult microglia perform effective and adequate responses, aging microglia no longer respond to stimuli in a correct manner. This altered response is driven by a particular signature that leads to the so-called senescent microglia and subsequently to dystrophic microglia. Dystrophic microglia may contribute to neurodegeneration.

Microglia from older mice at 24 months of age show a distinct genetic signature compared to their younger mice [55,56]. This gene pattern is linked to processes such as antigen presentation, phagosome and lysosome-related signaling pathways, oxidative phosphorylation, and others. This stands in contrast to the transcriptional profile of microglia undergoing acute activation, which primarily involves NF-κB signaling pathways not typically observed in aged microglia [55]. This suggests that in aged or long-lasting activated microglia, the functional state differs from that induced by an acute inflammatory response. Moreover, the expression of specific homeostatic genes, such as P2ry12 and Tmem119, is downregulated in aged microglia [55]. Activation of the NLRP3 inflammasome is also a significant contributor to the pro-inflammatory state of aged microglia and depletion of this factor diminishes the activation of microglia seen in aging, the increase in pro-inflammatory cytokine expression due to inflammation, and the memory decline [57,58].

In the aging brains of both humans and rodents, microglia store lipofuscin granules, which are large and indigestible compounds. These granules contain myelin fragments and unsaturated fatty acids, but also proteins and carbohydrates [59,60]. Moreover, recent studies [61] have revealed that approximately half of microglia in the aging brain accumulate lipid droplets. This subset of microglia has been termed lipid-droplet-accumulating microglia (LDAM) and has been found to exhibit a distinct transcriptome signature associated with increased release of CCL3, CXCL10, IL-6, heightened production of ROS and reactive nitrogen species (RNS) with impaired phagocytic activity [61]. Consequently, this phenotype can be considered indicative of dysfunctional microglia.

A key feature of aged microglia is their diminished ability to perform phagocytosis. A recent study in aged microglia highlighted the increased expression of CD22 as a negative regulator of phagocytosis [62]. Notably, inhibiting CD22 led to enhanced clearance of myelin debris or misfolded proteins such as amyloid-β oligomers and α-synuclein fibrils. This intervention also reversed the heightened expression of inflammatory factors and bolstered cognitive function in aged mice. These findings underscore the connection between age-related phagocytic impairment in microglial cells and cognitive decline.

Despite the stability in microglial cell numbers throughout aging in both humans and mice [63], microglia in the aging brain exhibit altered morphological and functional characteristics. In the elderly human brain [60], some microglia display a dystrophic morphology, marked by the loss of fine branches, the formation of cytoplasmic spheroids, fragmentation, or beading [64]. Notably, this severe form of morphological microglial aging is observed only in humans and not in rodents, likely due to differences in lifespan and environmental influences [65]. Nonetheless, microglia in aged rodents also undergo specific morphological changes indicative of a senescent phenotype [58,66]. These morphological changes result in a lower ability of individual microglia to scan territorial domains, reducing the brain volume covered by their processes. 

An analysis of gene expression patterns in microglia from aged humans and mice showed only a limited number of genes with significant overlap [29]. Several factors could explain this discrepancy, including differences in the lifespans of the species, the presence of diverse pathologies in human donors, and the increased complexity of microglial functions in the human central nervous system due to its greater overall complexity. Certain genes (e.g., ERAP1, ERAP2, and CD58) were found to be much less prevalent in mouse microglia with respect to human microglia [67]. During aging, many genes associated with the actin cytoskeleton, cell adhesion molecules, and cell surface receptors that are part of the microglial sensome (e.g., IL6R, P2RY12, and TLR10) showed reduced expression. Conversely, genes with increased expression in aging included IGF2R, CD163, the transcription factor RUNX3, integrin modulators DOCK1 and DOCK5, receptors CXCR4, and vascular endothelial growth factor A (VEGFA). The purinergic receptor P2RY12, which facilitates microglial chemotaxis, was also downregulated in aged human microglia, emphasizing changes in microglial sensing and movement with age. However, when comparing gene expression profiles of aged microglia between humans and rodents, both differences and significant overlaps are evident [67].

The so-called disease-associated microglia (DAM) are viewed as “defensive” microglia, whose primary role is to localize and clear damage [60] and are characterized by a decrease in the expression of genes linked to the homeostatic microglia, e.g., Cx3cr1 and purinergic receptors P2ry12/P2ry13 and Tmem119 [67,68]. DAM rises with age, comprising around 3% of all microglial cells in old mice (20 months). These cells are additionally distinguished by an increase in genes related to phagocytosis, lysosomal activity, and pathways involving lipid metabolism. The DAM phenotype is triggered by danger molecules found on apoptotic cell bodies, but also by myelin debris, a product of lipid breakdown and extracellular protein clumps. This process relies on TREM2 signaling and demonstrates the overlap between aging and CNS diseases (Section 3.6).

Understanding the aging-related changes in microglia is crucial for mitigating their detrimental effects and promoting healthy aging. However, the relationship between altered microglia and aging is complex. Altered microglia can both contribute to and result from the aging process. As already mentioned, microglia contribute to a state of chronic low-grade inflammation in the CNS, known as inflammaging. This persistent inflammatory state can accelerate the aging process and contribute to the development of age-related diseases, including neurodegenerative disorders. Pro-inflammatory microglial activity can damage neurons and other cells, leading to a decline in CNS function. As microglia age, their phagocytic efficiency declines, leading to the accumulation of cellular debris, dysfunctional proteins, and other waste products. This accumulation can create a toxic environment that promotes further cellular aging and dysfunction. Importantly, microglia are also involved in synaptic pruning and maintenance. Age-related changes in microglial function can disrupt synaptic homeostasis, leading to cognitive decline and other neurological deficits associated with aging. 

Aging affects the entire body, including systemic immune function. Changes in peripheral immune cells and circulating inflammatory mediators can influence microglial behavior and contribute to their altered state [69,70]. This systemic-to-CNS communication can exacerbate microglial dysfunction. Aging is also associated with widespread epigenetic changes, including in microglia. These changes can alter gene expression and cellular function, contributing to the aged phenotype of microglia. Epigenetic modifications can be both a cause and a consequence of cellular aging processes. Lastly, age-related increases in oxidative stress and mitochondrial dysfunction can affect microglial health and activity. These changes can lead to a feedback loop where damaged microglia further contribute to oxidative stress and inflammation, perpetuating the cycle of aging.

Olah et al. (2018) [56] used RNA sequencing (RNA-seq) to profile the gene expression of microglia isolated from the brains of elderly and young individuals. The data analysis confirmed how the microglia of elderly individuals show a significantly different transcriptional profile compared to young ones with an increase in the expression of pro-inflammatory genes and a decrease in anti-inflammatory genes, suggesting a state of chronic inflammatory disease that may contribute to inflammaging, a decrease in the expression of genes associated with phagocytosis and clearance of cellular debris, indicating a loss of function in maintaining brain homeostasis and alterations in energy metabolism. This study also identified potential biomarkers for early diagnosis of neurodegenerative diseases by performing proteomic analysis. Using mass spectrometry, the researchers profiled the proteins expressed in microglia and compared the protein profiles between young and elderly individuals, confirming the data obtained with transcriptional analysis [56]. The integration of transcriptional and proteomic data in this study offers a more comprehensive view of aging-associated changes in microglia. The results show that aged microglia not only have significant alterations in gene expression profiles but also exhibit changes in protein levels, reflecting an inflammatory profile and a reduced ability to maintain brain homeostasis.

### 3.6. Pathological (Neurodegenerative) Signature of Microglia (Figure 3)

Numerous microglial signatures have been identified as related to different pathologies from a transcriptional point of view. It is true that the aforementioned DAM can be considered a protective microglia but is more commonly associated with many pathologies.

In a model of AD, analysis of brain CD45+ cells through single-cell RNA-seq revealed three clusters of microglia [71]. One of these clusters was easily identified in age-matched wild-type animals. These cells downregulated genes associated with homeostatic microglia (TMEM119, AND P2RY12, and CX3CR1) and upregulated different genes (CLEC7A, ITGAX, CD9, and CD63), including those linked to AD risk factors (TREM2, DAP12, APOE, CTSD, and LPL). Among the two DAM clusters, one exhibited higher expression of these signature markers, which the authors interpreted to mean that the cluster with lower expression was an intermediate one [71]. Interestingly, the proposed intermediate signature was not dependent on TREM2, which was required for DAM development [71]. The intermediate signature was associated with a reduction in homeostatic genes and an increase in other genes linked to the signature (DAP12, APOE, CTSB, CTSD, B2M, LYZ2, and FTH1), whereas the DAM signature was linked to further upregulation of DAM genes. Putative DAM in situ, enriched for CD11c+, were found adjacent to plaques in the AD pathology mouse model; in humans, LPL+ microglia, also regarded as putative DAM, were found next to plaques in postmortem samples from individuals with AD. These cells were also found in the spinal cord of an ALS mouse model and showed increased prevalence with disease progression. All these data demonstrate a partial overlap between the aged signature and the pathological signature (DAM) of microglia, as well as a predictable overlap between the various microglial signatures characteristic of specific pathologies.

**Figure 3 ijms-25-10951-f003:**
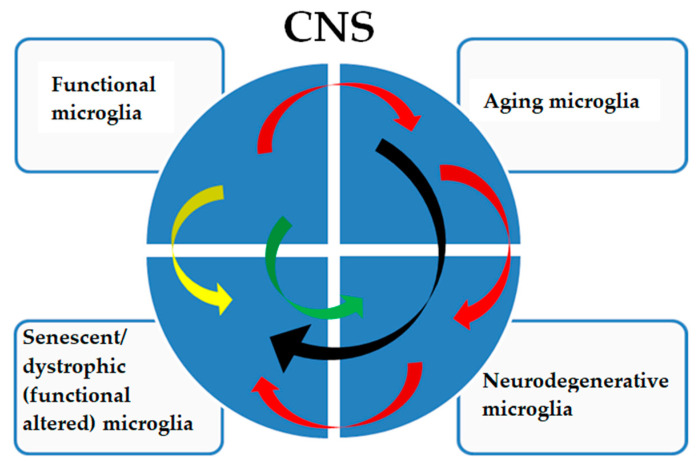
The pathological signatures of microglia show a complex relationship. Functional microglia in adults are subjected to physiological aging phenomena (red arrows), which can directly determine a senescent/dystrophic phenotype (black arrows) with a decreased ability to perform their functions and with cognitive and physical decline that is not caused by a neurodegenerative pathology. Aging microglia can also acquire a neurodegenerative signature (red arrows) with a frankly neurodegenerative phenotype, which will consequently determine a dystrophic phenotype (red arrows), worsening the pathology. Furthermore, functional microglia can acquire a senescent/dystrophic phenotype (yellow arrow), a prelude to non-aging neurodegenerative diseases or acquire a neurodegenerative signature such as, but not limited to, in type II microgliopathies (green arrow).

A specific microglial signature is characteristic of Nasu–Hakola disease, which is considered a type I microgliopathy. In this disease, functional microglia show functional alterations (Figure 3, yellow arrow) not fully correlated with a neurodegenerative or aged signature [72]. In type II microgliopathies, specific signatures, for example, allelic variants of TREM2, directly predispose (Figure 3, green arrow) to AD-like neurodegenerative phenotypes [73].

Many studies have shown that numerous genes linked to early-onset AD risk are predominantly expressed in microglia [73,74]. These genes include those encoding TREM2, ATP-binding cassette subfamily A member 7 (ABCA7) [75], members of the membrane-spanning 4-domains subfamily A (MS4A) 74, CD133 [76], the ε4 allele of APOE (APOE4) [77,78], PU.1 [79], and others. Moreover, the complement receptor CR1, expressed in microglia, has been linked with late-onset AD [80,81], reflecting the microglia’s involvement in phagocytosis of synapses and contributing to AD pathogenesis.

However, the pro-inflammatory signature in amyloid-β-plaque-associated microglia [67,82] was absent in microglia located in non-plaque areas, indicating that the neuroinflammation linked to AD primarily occurs in plaque-associated microglia [83]. Genes such as Axl, APOE, Clec7a, Itgax, Lgals3, and Cst7 are now recognized as part of a common neurodegenerative and DAM signature observed by multiple research groups. These studies highlight that microglial cellular reprogramming in response to neurodegeneration is related to unique underlying transcriptional programs. DAM microglia have also been detected in other conditions, including ALS, fronto-temporal dementia, and in a mouse model of severe neurodegeneration, suggesting that the DAM expression phenotype is generally triggered by disrupted brain homeostasis [67]. Although DAM microglia have been associated with amyloid-β plaques, it remains unclear whether they have a harmful or protective role. Innate immune cells are recognized for their role in mitigating AD by decreasing amyloid-β buildup in the initial stages of the condition [84]. However, as microglia age, their capacity to clear amyloid-β from the brain may diminish [85,86]. Furthermore, with the accumulation of amyloid-β, microglial cells tend to adopt a more neurotoxic phenotype [64,87,88], characterized by the expression of IL-1β, IL-6, TNF, and CCL3 [87]. The neurotoxicity of microglial cells may also be associated with their expression of tau, triggered by amyloid-β deposition in the brain, which is known to be neurotoxic [89].

The so-called microglial neurodegenerative phenotype (MGnD) is marked by the suppression of homeostatic genes (such as Cx3cr1, Olfm3, P2ry12, Csf1r, Tmem119, Tgfb1, Mafb, Mef2a, and Sall1) and the elevation of inflammatory genes (including APOE, Csf1, Itgax, and others) [90,91]. This genetic profile induces the activation of pathways that drive cytokine release, phagocytosis, and chemotaxis/migration, equipping MGnD to respond protectively to neuronal damage. The shift from homeostatic microglia to the MGnD state is controlled by the TREM2-APOE signal. Although MGnD and disease-associated microglia (DAM) share functional similarities, their transcriptional signatures are distinct [90].

MS is another inflammatory condition affecting the central nervous system (CNS), with microglia playing a significant role [92,93]. In the EAE mouse model of MS, microglia display a transcriptional profile that resembles those found in AD and ALS models [90,94,95]. This profile may drive disease progression. For example, microglia produce chemokines like Ccl2, which attract inflammatory CCR2+ monocytes from the bloodstream. Microglia uniquely express CX3CR1 in the brain, a receptor important for immunomodulation under normal conditions; its genetic removal leads to neurotoxicity [96]. Furthermore, decreased Cx3cr1 expression in microglia correlates with disease advancement in EAE [90]. This altered microglial phenotype in EAE also contributes to inflammation by activating astrocytes. In MS patients, microglia lose the P2RY12 marker, indicating a shift from a homeostatic to a pro-inflammatory state. They then express markers related to phagocytosis (e.g., macrosialin/CD68), antigen presentation (MHC class I and II molecules and CD86), and reactive oxygen species production (such as cytochrome b-245 light chain/CYBA) [97]. Minocycline, which targets microglia [98], showed positive effects in an MS clinical trial [99] but worsened ALS symptoms [100], possibly due to its varying impact on the microglial phenotype. This demonstrates that MS and ALS show similar but not identical signatures and minocycline may affect it in different manner. 

Many of the microglial alterations seen in ALS were also detected in a mouse model of AD, suggesting a shared microglial response across various neurodegenerative conditions. One of the initial descriptions of a DAM signature emerged from research on the SOD1 mouse model of ALS. In this model, microglia exhibited activation of pathways involving APOE, tau, and presenilin 2, with APOE being one of the most significantly upregulated transcripts during disease progression [101]. Similarly, a subsequent study also found enhanced APOE signaling in microglia in both human ALS cases and SOD1 mice. This study also found that the signature of homeostatic microglial disappeared as early as two months prior to detectable pathology onset in the SOD1 mouse model [102]. Studies have also uncovered mechanisms driving these microglial signature changes in ALS. ALS-related neuronal death activates TREM2-APOE signaling in microglia in response to apoptotic neurons. This signaling leads to the expression of miR-155, a known pro-inflammatory signal [90]. Genetic deletion of APOE was found to prevent the induction of miR-155 in microglia. Significantly, deleting miR-155 genetically re-established the homeostatic signature of microglia, prolonging SOD1 mice survival [102]. 

In addition to AD, other tauopathies such as progressive supranuclear palsy [103] and fronto-temporal dementia [104] present distinct microglial signatures. In these disorders, activated microglia contribute to the pathological accumulation of tau, a protein associated with neurofibrillary degeneration. Recent studies suggest that in progressive supranuclear palsy and fronto-temporal dementia, microglia express genes associated with chronic inflammation, such as TGFβ and IL-6, favoring the progression of pathological tau. This indicates a bidirectional relationship between inflammation and protein dysfunction.

In addition to the aforementioned TREM2-APOE pathways, microglia in ALS have been shown to exhibit a pro-inflammatory phenotype with the expression of genes such as secreted phosphoprotein 1, Ccl3, and Ccl4, which, like Ccl2, promote the infiltration of peripheral monocytes [105]. These genes are associated with increased disease progression and loss of microglia homeostatic function, which are implicated in inflammation-mediated neurotoxicity. An additional peculiarity of microglia in ALS is their increased expression of molecules involved in lipid metabolism, such as ABCA1 and APOE [105], suggesting a role for microglia in managing the altered lipid accumulation observed in these diseases.

A growing field concerns the role of microglia in Huntington’s disease, another neurodegenerative disease characterized by neuronal degeneration in the basal ganglia [106]. The microglial signature in HD includes the activation of inflammation-related genes such as IL-1β, TNF-α, and NLRP3, suggesting activation of the inflammasome pathway. This is particularly relevant because chronic inflammasome activation can accelerate neurodegeneration. In addition to inflammation, HD microglia show abnormalities in genes involved in the response to oxidative stress, such as HMOX1 (heme oxygenase 1), suggesting their impaired ability to handle the neurotoxic environment generated by huntingtin protein mutants [107].

In spinocerebellar ataxias, which involve neuronal degeneration of the cerebellum, microglia show alterations in the transcriptional profile affecting genes related to cell motility and migration [108,109], such as CX3CR1, in addition to the increase in molecules related to antigen presentation, such as MHC-II. This suggests that, in addition to inflammation, microglia in these diseases may contribute to the aberrant activation of the adaptive immune response, worsening neurodegeneration.

Recently, it has emerged that even diseases not classified as neurodegenerative, such as glaucoma, present a strong microglial activation [110]. The microglial signature in this context shows the expression of genes involved in the response to neuronal damage and inflammation, such as AIF1 (Iba1), together with pro-inflammatory molecules such as IL-6 and Ccl2. This highlights how microglia may contribute to optic nerve degeneration and suggests that modulating these cells could also have therapeutic benefits in ocular diseases.

### 3.7. Aging and Neurodegenerative Signature

To summarize, the signatures of aged and neurodegenerative microglia (DAM or MGnD) present both common and specific traits [51,55,80], with the latter being particularly relevant in neurodegenerative diseases typical of aging, such as AD, PD, and other dementias. Microglia play a central role in the brain’s immune response and their dysfunction is a key element in the pathogenesis of these diseases. The common traits between aged and pathological microglia are pro-inflammatory activation, in fact, both signatures show an increase in pro-inflammatory cytokines (IL-1β, TNF-α, and IL-6), which can contribute to chronic neuroinflammation and neurodegeneration and alteration in phagocytosis, with a reduced phagocytic capacity and reduced ability to remove cellular debris, abnormal proteins, such as amyloid-β and α-synuclein, and damaged neurons. Other common aspects are increased levels of ROS with oxidative damage to neurons and microglia themselves, and dysregulation of intracellular signaling, e.g., NF-κB and MAPK.

The two microglial signatures, aged and pathological (DAM/MGnD), also show specific traits [71]. Aged microglia can enter a state of senescence, characterized by a senescence-associated secretory profile [64,66], which releases inflammatory and pro-apoptotic factors and tends to show a reduction in motility and branching complexity, limiting its ability to monitor and respond to changes in the neuronal environment. It presents a reduction in pheno-typical plasticity by losing the ability to change between active and inactive states, negatively influencing their reactivity to injury or infection.

Pathological microglia exhibit aberrant, excessive, or dysfunctional reactivity, contributing to chronic neuroinflammation and neuronal degeneration. In diseases such as AD and PD, pathological microglia interact with pathological protein aggregates (amyloid-β, tau, and α-synuclein) [85,86], but often with reduced phagocytic efficacy, contributing to the accumulation of these aggregates. It also leads to an increase in the production of neurotoxic molecules such as NO and ROS, which can cause direct damage to neurons and worsen neurodegeneration. There are also alterations in lipid metabolism that influence cell membrane functions and intracellular signaling.

### 3.8. APOE Microglia Signature

Noteworthy is the relationship between the APOE and microglia (Figure 4). Lipids are gaining recognition as significant contributors to the balance of immune cells due to their various distributions within these cells. The distinct regions within the cell’s plasma membrane, known as lipid rafts, have the ability to initiate receptor interactions and signal transduction processes [111,112]. Conversely, lipids found within intracellular compartments, such as signaling messenger [113] or mitochondrial lipids [114], can modulate immune responses by influencing metabolic pathways. Lipid metabolism also plays a crucial role in controlling the state and function of microglia. For example, polyunsaturated fatty acids like those in the n-3 series, such as docosahexaenoic acid and eicosatetraenoic acid, have the capacity to shift microglia towards a phagocytic phenotype while reducing inflammatory markers through metabolic reprogramming [115,116,117].

APOE-mediated lipid metabolism also has an impact on microglial function.

APOE protein is involved in lipid transport in the human body and it has various genetic isoforms, APOE ε2, APOE ε3, and APOE ε4 [118]. APOE stands out as the most prominent risk factor associated with AD prevalence and the age at which the disease manifests itself and the presence of the APOE ε4 variant has been associated with an increased risk of AD developing. 

Two recent studies have investigated the relationship between microglia and APOE by complementary approaches. The first study employed gain-of-function using Cx3cr1creERT2/+ mice to induce the expression of APOE3 or APOE4 specifically in microglia and CNS-associated macrophages (CAMs) [119]. The second one, by loss-of-function approaches, using Cx3cr1creERT2/+ mice crossed with mice expressing floxed versions of APOE3 or APOE4, allowed for the deletion of APOE3 or APOE4 specifically in microglia and CAMs [120].

The study of Liu et al. (2023) [119], expressing APOE3 or APOE4 in microglia and CAMs of APP/PS1 mice, found that APOE3 expression increases the number of plaque-associated microglia and reduces insoluble amyloid levels, while APOE4 has opposite effects. Single-cell sequencing evidenced that APOE3 promotes a reactive and MGnD transcriptomic signature, whereas APOE4 induces a stress-related transcriptomic signature and mitochondrial dysfunction. Moreover, APOE3 expression reduced amyloid-β-associated cortical astrogliosis, whereas APOE4 promotes astrocytic activation in an amyloid-β-independent manner.

Similarly, in the second study [120], microglia from APOE4 knock-in mice showed increased expression of homeostatic genes compared to APOE3 mice, while in APOE3 knock-in mice exposed to apoptotic neurons, microglia mounted a robust phagocytic response and APOE4 microglia failed to induce this response. Deleting APOE4 in microglia rescued this response, enhancing the expression of disease-responsive genes. In AD models, microglial deletion of APOE4 reduced tau pathology and neuronal loss, restored microglial reactivity, and reduced amyloid pathology. Both studies concluded that APOE4 keeps microglia in a quiescent state, impairing their ability to respond protectively to AD pathology and exacerbating neurodegeneration.

Both aging and APOE4 are associated with increased brain inflammation. Microglia tend to shift into a pro-inflammatory state, releasing inflammatory cytokines such as IL-1β, TNF-α, and IL-6. The phagocytic capacity of microglia decreases with both age and the presence of APOE4. This impairs the ability of microglia to clear cellular debris, abnormal proteins, and apoptotic neurons, contributing to the accumulation of amyloid plaques and neurodegeneration. Both factors can increase levels of oxidative stress in microglia, leading to further cellular damage and neuronal dysfunction. Changes in intracellular signaling pathways are observed in both aging and APOE4 microglia, affecting their ability to respond appropriately to stimuli and maintain brain homeostasis.

APOE4 microglia show alterations in lipid metabolism, negatively affecting the functioning of cell membranes and intracellular signaling. APOE4 is associated with more neurotoxic microglia, which may contribute to neurodegeneration through the production of toxic substances and activation of pro-apoptotic pathways in neurons. APOE4 microglia show increased interaction with amyloid plaques but a reduced ability to phagocytose them, contributing to the accumulation of amyloid-β and the progression of AD.

These common and specific traits paint a complex picture of aging and APOE4-altered microglia, highlighting the importance of understanding their dynamics to develop targeted therapies for neurodegenerative diseases such as AD.

Krasemann et al. (2017) [90] disclosed a pivotal finding that links the acquisition of a neurodegeneration-associated phenotype by microglia to APOE and its mediation through TREM2 signaling. Apoptotic cells are effectively engulfed by microglia during nervous system development [121], leading to increased APOE expression, which is negatively associated with the downregulation of homeostatic genes [106]. Microglia in both developmental stages and neurodegenerative conditions may share similar regulatory mechanisms triggered by dying neurons. The authors proposed that MGnD microglia, amyloid plaques, and dystrophic neurites create a microenvironment rich in APOE, potentially playing a significant role in the progression of AD and other neurodegenerative disorders. 

In this context, the TREM2-APOE signaling pathway plays an important role in determining the transcriptional phenotype of dysfunctional microglia in disease, being central in the induction of the MGnD phenotype. TREM2 activation by APOE induces significant changes in microglia gene expression and in models of neurodegenerative diseases, such as APP-PS1 mice (a model for AD); the activation of this signaling pathway is associated with increased amyloid plaque pathology. TREM2 is a receptor expressed on the surface of microglia and plays a key role in regulating phagocytosis and the inflammatory response [122]. This activation occurs through the binding of specific ligands present on the surface of apoptotic neurons or released during neurodegeneration. Once activated, TREM2 transmits intracellular signals through the adaptor protein DAP12. This leads to the activation of various intracellular signaling pathways and transcription factors that modulate gene expression including APOE. APOE induction further promotes the MGnD phenotype, characterized by an increased inflammatory response and suppression of homeostatic genes. In addition, there is a feedback loop in which APOE can also influence TREM2 activity. APOE binds to damaged neurons and apoptotic debris, facilitating recognition and phagocytosis by TREM2. This feedback loop perpetuates TREM2 activation and APOE induction, contributing to the persistence of the neurodegenerative phenotype of microglia. The TREM2-APOE relationship can be considered a microglia-specific signature, especially in the context of neurodegenerative diseases. Indeed, it significantly influences the transcriptional profile of microglia, promoting a shift from a homeostatic state to a state associated with neurodegeneration. In models of neurodegenerative diseases, such as AD, microglia associated with amyloid plaques display an MGnD phenotype, characterized by activation of the TREM2-APOE pathway. APOE induction is closely related to TREM2 signaling and amyloid plaque accumulation, suggesting a key role in the pathogenesis of AD. The absence of APOE in phagocytic microglia may prevent the transition to the neurodegenerative phenotype and maintain a homeostatic state.

APOE signaling suppresses key transcription factors involved in the regulation of homeostatic microglia, such as PU.1, MEF2a, and SMAD3. This mechanism modulates the microglia phenotype through interaction with microglia-specific enhancers [84,90].

The TREM2-APOE signature not only characterizes the functional and transcriptional state of microglia but also plays a crucial role in the progression of neurodegenerative diseases, making it a potential target for targeted therapeutic interventions.

### 3.9. Specific Microglia Signature

Prion diseases, including Creutzfeldt–Jakob disease, are a group of rare but fatal neurodegenerative disorders characterized by the accumulation of prions in the brain. Previous studies have used mouse models of prion diseases and human brain samples affected by these pathologies, highlighting a characteristic signature of microglia. This is characterized by reduced expression of genes associated with phagocytosis, the anti-inflammatory response, and clearance of cellular debris. Microglia in brains affected by prion diseases show increased expression of pro-inflammatory genes that may contribute to the progression of neurodegeneration. The study also highlighted morphological alterations in microglia and modification of important signaling pathways including those linked to the TREM2 protein. The identified molecular pathways could represent potential therapeutic targets to modulate microglia activity and slow disease progression [123,124].

Microglial signatures show significant differences in their transcriptional, functional, and morphological profiles between males and females. RNA sequencing studies have shown that male and female microglia express different levels of specific genes involved in the immune response, phagocytosis, inflammation, and homeostasis. For example, pro-inflammatory genes tend to be more expressed in the microglia of males than females. Male and female microglia can respond differently to external stimuli, such as injury or infection. In general, microglia from males tend to show a more pronounced inflammatory response, whereas those from females may have a more modulated and protective response. Gender differences in microglia may contribute to variations in neurodegenerative disease risk and progression. In AD, female microglia, for example, maybe more efficient at clearing amyloid plaques than male microglia [125,126,127].

Physical exercise has neuroprotective and anti-inflammatory effects that influence the central nervous system, including microglia. Regular physical activity is associated with a reduction in the production of pro-inflammatory cytokines by microglia. This can contribute to a decrease in the inflammatory state in the brain, which is particularly useful in the context of neurodegenerative diseases such as AD and PD. Physical exercise can promote an anti-inflammatory and neuroprotective microglial phenotype. Microglia can express higher levels of genes associated with the production of anti-inflammatory cytokines and phagocytosis, improving the ability of microglia to remove cellular debris and misfolded proteins [128,129].

## 4. Conclusions

Microglia signatures refer to specific profiles of microglia activity or gene expression. Through a comprehensive analysis of gene and protein expression profiles, we can identify specific genes and proteins that characterize different states of microglial activation, including those associated with pro-inflammatory and anti-inflammatory states. This approach contributes to the knowledge base regarding the dynamics of microglial activation in both physiological and pathological conditions. The question of whether these signatures are a cause or a consequence of microglia-related brain disorders is highly relevant since understanding whether alterations in microglia are a primary cause of brain pathologies (e.g., neurodegenerative diseases such as AD) or whether they are a secondary response to such pathologies can significantly influence therapeutic strategies. Furthermore, understanding the role of the various microglial signatures could shed light on the most frequent “pathology”, including but not limited to CNS, which is aging.

We have already discussed how microglia can become hyperactive or dysfunctional, releasing inflammatory cytokines and neurotoxic factors that can damage neurons and the extracellular matrix. This may contribute to the pathogenesis of diseases such as AD and PD. Some genetic mutations, such as those in the TREM2 gene or the APOE4 isoform, can alter the function of microglia, making them more likely to cause brain damage. In these cases, dysfunctional microglia can be seen as a direct cause of the disease. The so-called microgliopathies [19] are an example of this.

Microglia can have a proactive role in shaping the neuronal environment. If altered, they can negatively affect synaptogenesis, synaptic plasticity, and clearance of cellular debris, leading to brain dysfunction. Microglia interact closely with neurons, astrocytes, and other cell types in the brain. Alterations in these interactions caused by primary diseases may lead to changes in microglia signatures as an adaptive response, which could exacerbate the disease.

Microglia may be activated in response to brain damage or other pathologies. In this scenario, alterations in their signatures could be a consequence of the presence of pathogens, abnormal protein deposits, or neuronal damage. In neurodegenerative disorders, neuroinflammation may be a secondary response to primary pathological processes. For example, in AD, microglia may be activated by amyloid-β deposits.

The relationship between microglia signatures and brain disorders is likely to be bidirectional and dynamic. In some conditions, dysfunctional microglia may actively contribute to disease onset and progression (cause), while in other situations, they may represent an adaptive or maladaptive response to pre-existing damage or pathology (consequence).

Understanding this complex interaction requires further research, including longitudinal studies and experimental models that can isolate the various factors involved. Unraveling these dynamics may lead to new and more effective therapeutic strategies for microglia-related brain diseases.

In this review, the main microglial signatures have been highlighted. It is clear that other and probably more complex signatures will be highlighted in the future.

Finally, if microglia signatures are a cause, preventive interventions or treatments that directly target microglia could be developed. If they are a consequence, therapies may need to focus on the primary causes of the disease, with microglia acting as markers of disease progression.

## 5. Future Perspectives

As techniques such as single-cell RNA seq and spatial transcriptomic analysis emerge, it is possible to obtain increasingly detailed profiles of different microglia subpopulations in various physiological and pathological states, such as in AD, MS, PD, and autism spectrum disorders. Identifying specific signatures for each condition will allow us to track the evolution of diseases over time and in specific brain areas.

Identifying unique signatures for “pro-inflammatory” or “protective” microglia could lead to the development of drugs that specifically modulate the behavior of microglia subpopulations. This could be crucial for neuroinflammatory diseases such as Alzheimer’s, where the balance between neuroprotective and neurotoxic microglia is altered. Therapies could aim to reduce toxic microglia and promote repair.

The idea of peripheral biomarkers, based on molecules released by microglia or modulated by their activity, to diagnose or monitor the course of brain diseases is also being developed. For example, specific signatures could be detected through blood or cerebrospinal fluid analyses, making diagnoses less invasive than current techniques. Furthermore, the use of brain organoids with microglia derived from human stem cells could provide more relevant models for the study of specific diseases and signatures. Precision medicine could integrate microglial signature profiles with other molecular factors to develop personalized therapies. For example, in patients with neurodegenerative diseases, a patient-specific characterization of microglia could direct treatment with drugs that modulate brain inflammation or neuronal damage.

Given the crucial role of microglia in synaptic plasticity and neuronal development, new avenues will open to study how microglial signatures may influence developmental disorders such as autism and schizophrenia. A better understanding of these dynamics could lead to early therapies to correct microglial abnormalities early in life. Finally, there is growing interest in developing immunotherapy approaches to modulate microglia activity. These could include monoclonal antibodies directed against specific microglial receptors or the use of drugs that alter expression signatures to make microglia more neuroprotective. Similarly, new imaging techniques, such as microglia-specific positron emission tomography, are set to improve, allowing microglial signatures to be visualized in real time in patients. This could revolutionize the monitoring of brain diseases, providing insights into the efficacy of therapies at the cellular level.

## Figures and Tables

**Figure 4 ijms-25-10951-f004:**
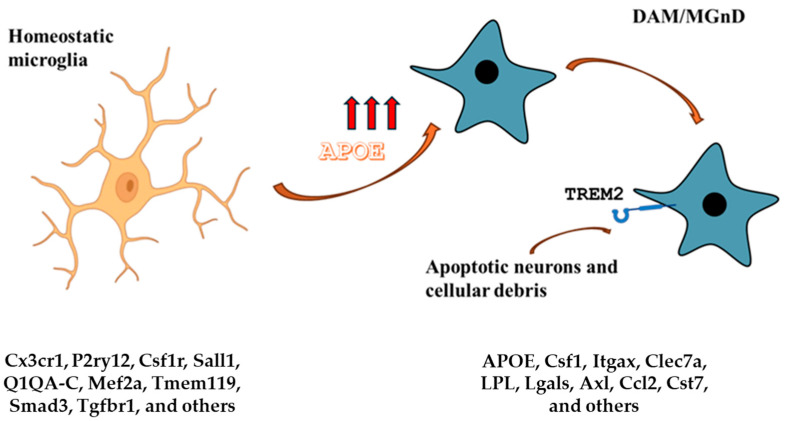
APOE and microglia. The homeostatic microglia signature is characterized by low APOE expression. On the other hand, DAM and MGnD signatures are characterized by high APOE expression and TREM2 signaling. The APOE-Trem2 axis regulates the microglia responses and in APOE or TRE2M knockout mice, the microglia response to brain tissue damage is severely blunted. The DAM signature is conserved in aging and neurodegenerative diseases and is triggered by apoptotic neurons and other cellular debris that accumulate in brain tissue damage. Gene expression markers for each condition are listed.

## Data Availability

No new data were created for this review.

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
