# Peer review of "Microglia Signatures: A Cause or Consequence of Microglia-Related Brain Disorders?"

_ijms, 2024, doi:10.3390/ijms252010951_

Round 1

Reviewer 1 Report

Comments and Suggestions for Authors

While the authors have provided a comprehensive overview of microglial signatures, I believe the article could be strengthened by incorporating more recent research. Many of the cited works date back to 2010-2020 or even older, and there have been significant advancements in the field within the past five years. Incorporating studies on microglial transcriptional signatures, single-cell gene sequencing, and transcriptomic analysis, related studies in the recent 5 years would enhance the article's relevance and provide a more up-to-date overview.

Minor concerns:

1.       The M1 label in Figure 1, it should be “inflammatory M1 microglia signature”.

2.       Section 3.7 has no references cited in the whole paragraph.

3. The authors may consider concluding the manuscript by discussing potential future directions in the field.

Comments on the Quality of English Language

Please review the document for typos and grammatical errors.

Reviewer 2 Report

Comments and Suggestions for Authors

In this work, the authors addressed various microglial signatures and their relationship with neurodegenerative and brain diseases, demonstrating that there is more complexity beyond the classical M1 and M2 phenotypes. Extensive and up-to-date literature was utilized to support this broad topic. Overall, the work is well-written and provides relevant information for researchers in the field. However, there are some issues that should be addressed by the authors to enhance their paper.

- Although the topic presented in this paper is highly relevant, there are several review papers that cover similar information, leading to some overlap with previous reviews. Therefore, I suggest that the authors clearly outline what is new and has not been previously reviewed in the literature to make the text more appealing to future readers.

- I suggest that the authors include more information about microglial phenotypes in neurodegenerative diseases, which impact millions of people worldwide.

- To assist readers, a list of abbreviations could be included.

- Some paragraphs are too brief and should be expanded upon or combined with other for better flow.

- Please provide additional keywords to help increase the discoverability of the paper.

- A significant flaw of the paper is the figures, which require substantial improvement. Currently, they do not accurately present the content of this review. The figures should also depict information on microglia markers during aging, diseases, development and other relevant features.
